# Identification of a βCD-Based Hyper-Branched Negatively Charged Polymer as HSV-2 and RSV Inhibitor

**DOI:** 10.3390/ijms23158701

**Published:** 2022-08-04

**Authors:** Rachele Francese, Claudio Cecone, Matteo Costantino, Gjylije Hoti, Pierangiola Bracco, David Lembo, Francesco Trotta

**Affiliations:** 1Laboratory of Molecular Virology and Antiviral Research, Department of Clinical and Biological Sciences, University of Turin, Regione Gonzole 10, Orbassano, 10043 Turin, Italy; 2Department of Chemistry, NIS Interdepartmental Centre, University of Turin, Via P. Giuria 7, 10125 Turin, Italy

**Keywords:** cyclodextrins, polymers, antiviral agents, herpesvirus, respiratory syncytial virus

## Abstract

Cyclodextrins and cyclodextrin derivatives were demonstrated to improve the antiviral potency of numerous drugs, but also to be endowed with intrinsic antiviral action. They are suitable building blocks for the synthesis of functionalized polymer structures with potential antiviral activity. Accordingly, four water-soluble hyper-branched beta cyclodextrin (βCD)-based anionic polymers were screened against herpes simplex virus (HSV-2), respiratory syncytial virus (RSV), rotavirus (HRoV), and influenza virus (FluVA). They were characterized by FTIR-ATR, TGA, elemental analyses, zeta-potential measurements, and potentiometric titrations, while the antiviral activity was investigated with specific in vitro assays. The polymer with the highest negative charge, pyromellitic dianhydride-linked polymer (P_PMDA), showed significant antiviral action against RSV and HSV-2, by inactivating RSV free particles and by altering HSV-2 binding to the cell. The polymer fraction with the highest molecular weight showed the strongest antiviral activity and both P_PMDA and its active fractions were not toxic for cells. Our results suggest that the polymer virucidal activity against RSV can be exploited to produce new antiviral materials to counteract the virus dissemination through the air or direct contact. Additionally, the strong HSV-2 binding inhibition along with the water solubility of P_PMDA and the acyclovir complexation potential of βCD are attractive features for developing new therapeutic topical options against genital HSV-2 infection.

## 1. Introduction

Cyclodextrins (CDs) are truncated cone-shaped oligosaccharides composed of α (1→4)-linked glucopyranose units arranged surrounding a slightly lipophilic inner cavity. Alpha (α), beta (β), and gamma (γ) cyclodextrin (CD) are the most common products present on the market, characterized by six, seven, and eight glucopyranose units, respectively [1,2]. The presence of this peculiar domain enables the latter to accommodate guest molecules through the formation of inclusion complexes with both hydrophobic molecules, but also hydrophilic moieties and ions [3,4]. Furthermore, the high number of hydroxyl functionalities makes CDs suitable building blocks for the synthesis of polymer materials [5,6,7]. Both water-soluble (hyper-branched) and water-insoluble (cross-linked) structures have been obtained [8,9] through the choice of proper synthetic conditions and the exploitation of, e.g., di-, tri-, or tetra-functional molecules as cross-linkers. Considering these characteristics, some applications of CD-based polymers include, e.g., (i) increasing the solubility of poorly soluble drugs, (ii) modulating drug release and activity, (iii) the protection of drugs from degradation, (iv) enhancing bioactivities, (v) the adsorption of undesired substances, and (vi) gene delivery [10,11,12,13,14]. In the field of virology, CDs have been recently proposed as potential antiviral agents, both as drug delivery systems and molecules with intrinsic antiviral action [15].

CDs and CD derivatives have been demonstrated to efficiently improve solubility, bioavailability, and antiviral potency of numerous antiviral drugs such as the anti-herpetic acyclovir and ganciclovir, and the antiretroviral azidothymidine, efavirenz, and darunavir [15,16,17,18,19,20,21,22,23], indicating them as flexible and resourceful compounds for formulation design [21]. On the other hand, numerous studies revealed that CDs are not as inert as believed for many decades showing their intrinsic antiviral activity against several common viruses [15,24]. The most frequently reported CDs and CD derivative mechanisms of action include the virucidal activity and the inhibition of the early steps of virus replication. Notably, the virucidal activity was associated with the ability of CD to complex the cholesterol molecules of the host cell membrane or the viral lipid envelope, respectively, rendering the cell less susceptible to viral infections or causing a direct impairment of the viral particle [24,25,26,27,28,29,30]. Alternatively, Jones et al. recently reported the strong virucidal activity of CDs modified with mercaptoundecane sulfonic acids due to their mimicking action of key cellular receptors [31]. A CD-mediated inhibition of virus binding to the host cell was also observed in several cases, in which CD derivatives, containing positively or negatively charged moieties, interacted with viral or cellular receptors, thus avoiding virus binding and entry into the host cell [32,33,34,35,36,37].

All these studies point out the strong antiviral potential of CDs and CD derivatives and the possibility to combine their widely demonstrated properties of complexation, drug delivery, and receptor blocking.

In this context, we synthesized and characterized a library of four different βCD-based polymers, by exploiting different synthetic procedures. The approaches ranged from consolidated reactions performed in organic solvents, reactions performed in water media aiming to avoid the presence of fossil-based solvents, and reactions performed in natural deep eutectic solvents (NADES) as a novel sustainable synthetic approach. All the syntheses were addressed to obtain water-soluble polymers, characterized by the presence of negatively charged functions, which would confer the ability to bind the positively charged domains of viral receptors with a consequent infectivity inhibition. Our research explored the intrinsic antiviral potential of these polymers against four viruses, namely, herpes simplex virus type 2 (HSV-2), human respiratory syncytial virus (RSV), human rotavirus (HRoV), and influenza virus type A (FluVA), chosen as representative of different virus characteristics, such as the presence or absence of a lipid envelope, a DNA or RNA genome, and heparan sulfate proteoglycan (HSPG) dependency for virus attachment [38,39,40,41]. Herein, this study reports the identification of a βCD-based polymer, named P_PMDA as a novel potent HSV-2 and RSV inhibitor, and it preliminary investigates the mechanism of P_PMDA antiviral action.

## 2. Results and Discussion

### 2.1. Material Design and Characterization

With the purpose of synthesizing negatively charged βCD-based water-soluble polymers, pyromellitic dianhydride, citric acid, carbonyl diimidazole, and 1,4 butanediol diglycidyl ether were exploited as the linking agents to obtain hyper-branched structure. On the other hand, isethionic acid and sodium citrate were chosen to impart negative charges to the resulting products in those cases in which the linking agent was not sufficient to achieve this target feature alone, as reported in Figure 1 (and Appendix A). The selected molar ratios between the reactants and the introduction of functional molecules, as in the case of isethionic acid and sodium citrate, were also exploited and optimized to achieve better control over the polymerization reactions and to hinder the formation of a cross-linked structure.

In Figure 2, the attenuated total reflectance Fourier transform infrared (FTIR-ATR) spectra acquired on the polymer products are reported. As a confirmation of the mechanisms involved in the formation of the polymer structure, the FTIR-ATR spectrum of P_PMDA (Figure 2A) is characterized by a carbonyl signal (C=O stretching) centered at 1722 cm^−1^ and not at 1769 cm^−1^, as it is observed for pyromellitic dianhydride (Appendix A). This feature together with the absence of the absorption at 1850 cm^−1^, characteristic of anhydride rings (Figure 2A) indicate that the latter underwent a ring-opening reaction during the synthesis. Moreover, the presence of a broad absorption centered at 1570 cm^−1^, typical of carboxylates, indicates that the carboxylate pendants, as displayed in Figure 1A, are generated from the pyromellitic dianhydride’s carboxyl groups which are not directly involved in the formation of the polymer chains, probably associated with triethylammonium molecules as cations. Similarly, citric acid/levulinic acid NADES-derived polymer (P_CA_LA) and carbonyl diimidazole-linked polymer (P_CDI_ISE) (Figure 2B,C) are characterized by a carbonyl signal (C=O stretching) centered at 1713 cm^−1^ and 1744 cm^−1^, respectively, corresponding to the formation, during the polymer synthesis, of carbonate bridges as a consequence of the nucleophilic substitution of carbonyl diimidazole’s carbonyl, and ester bridges generated via esterification reaction through anhydride intermediate formation typical of citric acid catalyzed by sodium hypophosphite (Figure 1B,C). Additionally, in the case of P_CDI_ISE, further proof is given by the elemental analysis of the product which confirms the presence of sulfur atoms, equal to 6.11±0.16 wt%, composing the polymer structure (Table 1). The FTIR-ATR spectrum of P_BDE_CIT (Figure 2D) gives information related to the presence of the carboxylate pendants imparted by sodium citrate molecules (Figure 1D), associated with a broad absorption centered at 1582 cm^−1^. Differently, the signals related to the ether bridges formed during the ring-opening reaction of 1,4 butanediol diglycidyl ether’s epoxy rings are overlayed with the glycosidic bond characteristic of the βCD (Figure 2D).

To better evaluate the presence of negative charges within the structure of polymers, zeta (ζ)-potential measurements were performed by preparing a dispersion of each product in ethanol, at room temperature. As a result, P_PMDA displayed the highest negative charge with a ζ-potential equal to −31.4 ± 1.7 mV, followed by P_CDI_ISE with a ζ-potential equal to −17.5 ± 0.6 mV, P_CA_LA with a ζ-potential equal to −10.6 ± 1.1 mV, and eventually by P_BDE_CIT with a ζ-potential close to zero (Table 1).

In addition, potentiometric titrations were performed (Appendix A) to confirm the results obtained from the ζ-potential measurements. The results are expressed as milliequivalents (m_eq_) of acidity per 100 g of the sample tested, thus proportional to the number of functionalities displayed by each polymer. As previously observed, P_PMDA resulted in the polymer with the highest number of negative charges, displaying 872 ± 10 m_eq_ of acidity, followed by P_CA_LA which was characterized by 755 ± 9 m_eq_ of acidity. In agreement with the ζ-potential measurements, P_CDI_ISE and P_BDE_CIT were characterized by a lower number of negative charges, displaying 80 ± 9 m_eq_ and 33 ± 7 m_eq_ of acidity, respectively (Table 1).

The thermal stability of the samples was studied via thermogravimetric analysis (TGA) and reported in Appendix A. The profiles are characterized by a first weight loss phenomena taking place between 50 °C and 100 °C, which was related to the volatilization of the water adsorbed on the surface of the polymers, being the latter hydrophilic because of the nature of the chosen building block. Subsequently, a two-step degradation process is observed between approximately 200 °C and 400 °C, associated with a competition between intramolecular and intermolecular paths, leading to the production of volatile products and a carbon residue, which further displayed progressive volatilization between 400 °C and 700 °C, giving a final carbon residue comprised between 10% to 20% of the initial weight. P_BDE_CIT resulted in the product characterized by the highest thermal stability (Table 1), displaying T_onset_ equal to 270 °C, followed by P_CDI_ISE (T_onset_ of 240 °C), P_PMDA (T_onset_ of 165 °C), and P_CA_LA (T_onset_ of 160 °C). The different thermal behaviors observed are strictly related to the stability of the chemical bonds characterizing the polymer network. As a confirmation, ether bonds (P_BDE_CIT) are more stable if compared to carbonate bridges (P_CDI_ISE), which are more stable if compared to ester bridges (P_PMDA and P_CA_LA).

The polymer design was directed at obtaining CD-based hyper-branched polymer structures functionalized with negative charges such as carboxylates and sulphonate groups to study if the demonstrated antiviral activity of βCD can also be extended to more complex molecular architectures [15,23,24]. Indeed, polyanionic compounds have been largely investigated for their ability to directly bind the positively charged basic domains of the determinants of virus infectivity [42,43,44]. They may interact with cell-free viral particles determining a complete virus inactivation [31] or act at the cellular level, exerting a competitive action with cellular receptors and avoiding virus attachment to cell [43]. Viruses exploiting the negatively charged HSPGs on the host cell surface for their first step of cell attachment (such as RSV, HIV, HSV-2) [45,46] are considered the major target of polyanionic compounds, which can act as HSPG antagonists, blocking virus–cell first interactions [43]. Moreover, the presence of βCD, as the polymer building block, may further increase the polymers’ antiviral effect, exploiting the possibility to load antiviral drugs as host–guest complex within the βCD units [23]. In addition, the synthesis of soluble polymer materials allows us to screen their processing from water solutions into, e.g., fibers via electrospinning technique or particles via nano spray drying technique. This feature offers the possibility to obtain, in the first case, fabrics composed of an antiviral matrix, while in the second case the approach can be exploited to achieve the deposition of an antiviral coating over target surfaces [47,48].

### 2.2. Antiviral Activity of the βCD-Based Hyper-Branched Polymers

In vitro antiviral assays were performed to assess the antiviral potency of the library of negatively charged polymers against RSV, HSV-2, HRoV, and FluVA by means of virus inhibition assays. Among all the tested polymers, P_PMDA was the most effective. Indeed, we demonstrated that it was significantly active against RSV, HSV-2, and FluVA, with EC_50_s values of 9.72 µg/mL, 0.18 µg/mL, and 61.40 µg/mL, respectively (Table 2). P_CA_LA showed a strong to moderate inhibitory activity against HSV-2 and RSV (EC_50_ of 4.12 µg/mL and 33.29 µg/mL, respectively), while P_CDI_ISE was only slightly active against RSV (EC_50_: 326.6 µg/mL). By assuming a dependency between the antiviral activity and the presence of negative charges characterizing the polymer structures, what observed is consistent with the results obtained from both the ζ-potential measurements and potentiometric titrations. As proof of this concept, P_PMDA, which resulted in the most effective polymer within the screened library, displays the highest negative charge. On the contrary, P_BDE_CIT is characterized by a negligible charge and consequently its antiviral effect was negligible. P_CDI_ISE and P_CA_LA, which are characterized by an intermediate number of charges, resulted in mild to moderate inhibitory activities.

As expected, the entire library of compounds was ineffective against HRoV (Table 2). The lack of anti-HRoV activity along with the predominant action of P_PMDA against RSV and HSV-2 can be explained by the dependence of these two latter viruses on the highly sulfated HSPGs for their attachment to cells. Due to its high number of negative charges, P_PMDA can efficiently alter the electrostatic interaction between HSPGs and HSV-2 and RSV surface proteins [49,50], actually acting as decoy receptors. On the contrary, since HRoV entry is a complex multistep process that does not involve HSPGs [39,51], no antiviral action against this pathogen was observed, a result that strengthened our previous hypothesis. The moderate activity against FluVA may be due to a partial electrostatic interference of P_PMDA with essential interactions between FluVA receptor (hemagglutinin) and the negative sialic acid residues that FluVA exploits as a cell receptor [52]. However, since we observed an antiviral activity mainly addressed to enveloped viruses (i.e., HSV-2, RSV, and FluVA), we cannot exclude a partial contribution of the previously described cholesterol complexation ability of βCD [24]. Our βCD-based polymers, although they are complex and hyper-branched structures, can still present free and active βCD cavities which can complex and deplete cholesterol from the viral envelope, causing a structural damage.

Further studies on the mechanism of action addressed this point. To exclude the possibility that the observed antiviral activity was due to a cytotoxic effect, viability assays were performed. The active compounds were not toxic for cells, showing CC_50_s > 3000 µg/mL for all the different cell lines. This result was likely related, firstly, to the biocompatibility of the chosen building block [3,4,24], and secondly, to the purification steps aimed to separate the polymer from undesired non-reacted reagents and by-products. Previous studies showed that some βCD-based polymers did not induce cytotoxicity and hemolysis in vitro [53]. Acute and repeated dose toxicity of βCD-based polymers have been tested on mice by Shende et al. The polymers were found to be safe and with good biocompatibility after oral administration at selected doses [12]. Additionally, Rassu et al. observed low toxicity and good cell uptake of βCD-based polymers studied as excipients to improve the nasal drug absorption [12,54].

P_PMDA showed highly favorable selectivity indexes when tested against HSV-2 and RSV. The synthesis of the polymer has been optimized by the work of Trotta et al., where the capability to generate electrostatic interactions with metal cations was demonstrated [55]. Furthermore, the same synthesis carried out using αCD instead of βCD was used to develop suitable nanocarriers for oxygen storage and release, aimed to limit hypoxia–reoxygenation injury in a cardiac cell model [56]. Therefore, this well-characterized polymer and the abovementioned viruses were chosen for further investigations.

### 2.3. Identification of the Active P_PMDA Molecular Fraction

As reported by Bianculli et al. and Kalitnik et al., the antiviral activity displayed by polysaccharides is associated with their molecular weight [57,58]. It has also been generally observed that the antiviral activity of sulphated polysaccharides is proportional to increasing molecular weight values [59]. Consequently, to investigate the antiviral effect of P_PMDA as a function of its molecular weight, different molecular weight fractions were obtained via the ultrafiltration technique.

The obtained P_PMDA fractions were screened against HSV-2 and RSV by means of the virus inhibition assays described in the Materials and Methods section. Results indicated that the P_PMDA fractions with the highest molecular weight (P_PMDA_50_ and P_PMDA_30/50_) were the most effective fractions against RSV, with EC_50_ values comparable to the ones obtained with the complete preparation (Table 3). In the case of HSV-2, these two fractions were significantly more active than the original compound, with improved EC_50_s and SIs. Since P_PMDA_50_ was the active fraction with the highest SI, it was selected for the next mechanism of action studies. Notably, the treatment with P_PMDA_10/30_ determined statistically higher EC_50_ values compared to P_PMDA for both RSV and HSV-2 (Table 3), suggesting that this fraction only partially contributes to the overall antiviral activity of the original unfractionated compound. The higher antiviral activity of P_PMDA_50_ and P_PMDA_30/50_ compared to P_PMDA_10/30_ may be due to the capability of larger polymer chains surrounding the viral particles by means of multiple interactions, therefore resulting in a complete masking of viral receptors and a maximized antiviral effect. On the contrary, shorter polymer structures are unable to form strong enough interactions and are therefore less effective.

### 2.4. Study of the Antiviral Mechanism of Action of P_PMDA and Its Active Fraction

The third set of antiviral experiments was addressed to investigate the mechanism of antiviral action of P_PMDA and P_PMDA_50_. Taking into account the structural and chemical nature of these compounds and the experimental conditions whereby they were initially tested, we hypothesized two different mechanisms of action: the direct impairment of viral particle or the alteration of virus binding to the host cell. Therefore, we first investigated whether P_PMDA and its active fraction were endowed with virucidal activity by performing virus inactivation assays. Both the complete preparation and the fraction showed significant virucidal activities against RSV particles (Figure 3B and Appendix A), reaching nearly 1 Log of viral titer inhibition at each tested concentration (500 µg/mL and 300 µg/mL). Both compounds were also able to partially inactivate HSV-2 (Figure 3A and Appendix A, and Appendix A), reaching the strongest statistical inhibition at 15 µg/mL. Nevertheless, the virucidal activity against HSV-2 was not sufficient to justify the strong antiviral action observed in the initial standard antiviral assay. Results from the binding assays (Figure 3C and Appendix A) demonstrated that the polymers strongly reduce the titer of HSV-2 bound to cells, suggesting that binding inhibition is the main mechanism of action against this virus. In particular, P_PMDA showed a superior effect compared to its fraction, reaching a higher Log of inhibitions (Appendix A). On the contrary, P_PMDA and P_PMDA_50_ only partially altered RSV binding to cells (Figure 3D), showing inhibition Log < 1 at all the tested concentrations (Appendix A) (Additional tested doses and numerical results of both the inactivation and binding assays are reported in the Appendix A).

Interestingly, P_PMDA showed different mechanisms of action against the tested viruses and a stronger activity against HSV-2. This was not surprising considering the widely different structural and replicative nature of these pathogens [38,40]. In view of our results, we hypothesized that P_PMDA can strongly interact with free RSV particles, completely surrounding and inactivating the virions with strong and irreversible interactions. The lack of antiviral activity during the binding test, in which the compound interacts with the virus at the cellular level for 1 h at 4 °C, suggests that a low temperature is not appropriate to facilitate these strong interactions. On the other hand, the binding inhibition of HSV-2 particles at low concentrations of P_PMDA indicated that the link between P_PMDA and HSV-2 is presumably transient and not able to permanently inactivate the viral particles before cell infection. The simultaneous presence at the cell surface level of P_PMDA and HSV-2 is necessary for the antiviral activity to occur.

To the best of our knowledge, few reports have described the anti-RSV activity of CDs or CD derivatives. Jones et al. recently synthesized mercaptoundecane sulfonic acid-modified CDs to mimic heparan sulfates and provide a nontoxic virucidal action. These CD derivatives were active against multiple viruses, in particular against RSV and HSV-2, showing a significant virucidal activity [31]. This is in line with our results, showing net virucidal activity of P_PMDA against RSV and strengthening the concept that CDs are suitable building blocks for the development of new antiviral strategies against this pathogen. Despite the use of P_PMDA to treat RSV systemic infections may be considered unlikely, we propose the potential use of this polymer for the synthesis of new antiviral materials, such as antiviral coatings for surfaces or daily used objects, which would help reduce viral dissemination. Indeed, RSV is spread through contact with droplets from infected people and through self-contamination after touching dried respiratory secretions on hard surfaces where the virus can remain for several hours [60,61]. The virucidal potential of P_PMDA, when employed to produce antiviral materials, needs to be further investigated.

With regards to HSV-2, different studies report the efficient delivery of acyclovir using βCD [16,22,62,63,64,65]. Nevertheless, the intrinsic anti-HSV-2 action of CDs or CD derivatives is less reported. Herein, we showed that P_PMDA is endowed with significant binding inhibitory activity against HSV-2. This characteristic along with the water solubility of P_PMDA and the acyclovir complexation potential of βCD represent a good starting point for the development of a new therapeutic topical option to treat genital HSV-2 infection. Additional studies are necessary to verify the purity of the polymer, such as the absence of unreacted and potentially allergenic pyromellitic dianhydride, together with the biocompatibility, the safety and the antiviral activity of P_PMDA in preclinical in vivo models.

## 3. Materials and Methods

### 3.1. Materials

β-cyclodextrin (βCD) was provided by Roquette Freres (Lestrem, France) and dimethyl sulfoxide (DMSO), N, N-dimethyl formamide (DMF), triethylamine, pyromellitic dianhydride, carbonyl diimidazole, 1,4 butanediol diglycidyl ether, isethionic acid, citric acid, sodium citrate, sodium hydroxide (NaOH), hydrochloric acid (HCl), oxalic acid, and sodium hypophosphite monohydrate were purchased from Sigma-Aldrich (Darmstadt, Germany). Levulinic acid was purchased from Acros Organics (Geel, Belgium). βCD was dried in an oven at 75 °C up to constant weight before use.

### 3.2. Polymer Synthesis

#### 3.2.1. Pyromellitic Dianhydride-Linked Polymer (P_PMDA)

The polymer was synthesized following the procedure described in a previous work [55]. Briefly, 0.98 g (0.86 × 10^−3^ mol) of anhydrous βCD was solubilized in 6 mL of DMSO. Afterwards, 1.00 mL (7.17 × 10^−3^ mol) of TEA was added under vigorous stirring at room temperature, followed by 2.25 g (10.33 × 10^−3^ mol) of pyromellitic dianhydride. The reaction was then allowed to develop for 24 h under stirring, at room temperature. An increase in the viscosity was observed during that time, as proof of the occurrence of the polymerization reaction. At the end of the reaction, the product was firstly precipitated, washed with ethyl acetate, then recovered by vacuum filtration. Subsequently, the dry product was solubilized in distilled water and purified via the ultrafiltration technique using a membrane with a cut-off of 10 kDa. The polymer was then recovered from the ultrafiltration cell and subsequently freeze-dried, obtaining a white powder as the product.

In addition, to evaluate the antiviral effect as a function of the molecular weight of the polymer, different molecular weight fractions were collected from P_PMDA by solubilizing the polymer in deionized water and by filtering the so-obtained solution via the ultrafiltration technique with different membranes, characterized by molecular weight cut-off equal to 10 kDa, 30 kDa, and 50 kDa. Each fraction was subsequently freeze-dried to obtain a dry powder. Following this procedure, four different molecular weight fractions of P_PMDA were collected: (i) not fractionated (P_PMDA_tot_), (ii) molecular weight higher than 50 kDa (P_PMDA_50_), (iii) molecular weight between 30 kDa and 50 kDa (P_PMDA_30/50_), and (iv) molecular weight between 10 kDa and 30 kDa (P_PMDA_10/30_).

#### 3.2.2. Citric Acid/Levulinic Acid NADES-Derived Polymer (P_CA_LA)

The synthesis of the polymer was carried out by dissolving 2.50 g (2.20 × 10^−3^ mol) of anhydrous βCD and 1.25 g (1.18 × 10^−2^ mol) of sodium hypophosphite monohydrate as the catalyst, into 10.00 g of a NADES composed of citric acid and levulinic acid at the weight ratio of 1:1, preheated at 140 °C. The reaction was carried out under vacuum by means of a diaphragm pump and under stirring for 4 h at 110 °C, by means of a hotplate stirrer. At the end of the reaction, the product appears as a yellow bulk solid. Later, the dry product was solubilized in distilled water and purified via the ultrafiltration technique using a membrane with a cut-off of 10 kDa. The polymer was then recovered from the ultrafiltration cell and subsequently freeze-dried, obtaining a white powder as the product.

#### 3.2.3. Carbonyl Diimidazole-Linked Polymer (P_CDI_ISE)

The synthesis of the polymer was carried out by dissolving at room temperature, 1.00 g (8.81 × 10^−4^ mol) of anhydrous βCD in 7.5 mL of DMF, using a round-bottom flask, under stirring. After complete solubilization, 1.00 g (7.93 × 10^−3^ mol) of isethionic acid was added, followed by 1.14 g (7.05 × 10^−3^ mol) of carbonyl diimidazole. After a transparent liquid was obtained, the solution was heated up to 90 °C using a hotplate stirrer equipped with thermoregulation and a metal hemispheric bowl to obtain a homogeneous heating of the flask. The reaction was then allowed to develop for 120 min under stirring; an increase in the viscosity was observed during that time, as proof of the occurrence of the polymerization reaction. Afterwards, the product was precipitated and purified with acetone, then recovered by vacuum filtration. Later, the dry product was solubilized in distilled water and purified via the ultrafiltration technique using a membrane with a cut-off of 10 kDa. The polymer was then recovered from the ultrafiltration cell and subsequently freeze-dried, obtaining a white powder as the product.

#### 3.2.4. 1,4 Butanediol Diglycidyl Ether-Linked Polymer (P_BDE_CIT)

The synthesis of the polymer was carried out by dissolving in a round-bottom flask, 1.00 g (8.81 × 10^−4^ mol) of anhydrous βCD in 5 mL of 0.2 M NaOH distilled water solution, using a round-bottom flask. Thereafter, 134 mg of sodium citrate (0.52 × 10^−3^ mol) was added. Eventually, 300 µL of 1,4 butanediol diglycidyl ether was added and the temperature was increased to 90 °C, using a hotplate stirrer equipped with thermoregulation and a metal hemispheric bowl to obtain homogeneous heating of the flask. The reaction was then allowed to occur for 90 min under stirring; an increase in the viscosity was observed during that time, as proof of the occurrence of the polymerization reaction. Afterwards, the pH of the solution was adjusted to 7 using HCl and the polymer was purified via the ultrafiltration technique using a membrane with a cut-off 10 kDa. The product was then recovered from the ultrafiltration cell and freeze-dried, obtaining a white powder as the product.

### 3.3. TGA Characterization

Thermogravimetric analyses (TGA) were carried out using a TA Instruments Q500 TGA (New Castle, DE, USA), from 50 °C to 700 °C, under nitrogen flow, with a heating rate of 10 °C/min.

### 3.4. FTIR-ATR Analysis

A Perkin Elmer Spectrum 100 FT-IR Spectrometer (Waltham, MA, USA) equipped with a Universal ATR Sampling Accessory was used for FTIR-ATR (Fourier transform infrared spectra-attenuated total reflection) characterization. All the spectra were collected in the wavenumber range of 650–4000 cm^−1^, at room temperature, with a resolution of 4 cm^−1^ and 8 scans/spectrum.

### 3.5. Elemental Analysis Characterization

The samples’ chemical composition was studied using a Thermo Fisher FlashEA 1112 Series elemental analyzer (Waltham, MA, USA).

### 3.6. ζ-Potential Analysis

A Malvern Zetasizer Nano–ZS (Malvern, UK) was used to measure the ζ-potential. All the tests were performed using ethanol at room temperature.

### 3.7. Potentiometric Titration

The content of negatively charged functionalities on the polymer products was determined via potentiometric titration, according to the procedure described by Soto et al., with slight modifications [66]. A 0.10 M NaOH solution was used as a titrant, pre-standardized with a 25 mM oxalic acid solution. The titration was carried out by adding 0.1–0.5 mL of the titrant each step, by means of a volumetric burette, into the sample solution which was kept under gentle stirring and at room temperature. The titrant additions were performed 60 s delayed one to each other, to allow the equilibrium to be reached. Moreover, pH values were continuously measured and recorded, by means of a pH meter, after each addition, until the pH of 12 was reached. Subsequently, the titration curve of pH versus titrant volume was generated and the curve’s inflection point was found via the second derivative approach. The volume of NaOH consumed at the inflection point was applied to Equation (1), and the milliequivalents of acidity per 100 g of sample were calculated as follows:m_eq_ of acidity/100 g sample = ((V_s_−V_b_) mL × c_NaOH_ × 100)/m_s_)(1)
where m_eq_ are milliequivalents, V_s_ and V_b_ are the volumes of NaOH consumed by the sample and the blank (β-CD was considered as a blank), whereas c_NaOH_ is the concentration of NaOH in mol/L, and *m_s_* is the mass of the sample. The potentiometric titrations were performed in duplicate.

### 3.8. Cell Lines and Viruses

Human epithelial cells (Hep-2) (ATCC^®^ CCL-23), human lung carcinoma epithelial cells (A549) (ATCC ^®^ CCL-185), African green monkey kidney epithelial cells (MA104) (ATCC^®^ CRL-2378.1), and African green monkey fibroblastoid kidney cells (Vero) (ATCC CCL-81) were grown as a monolayer in Dulbecco’s modified Eagle’s medium (DMEM; Sigma-Aldrich) supplemented with 10% FBS (and 1% Glutamax-I—Invitrogen, Carlsbad, CA, USA—in the case of MA104 and A549 cell lines). The media were supplemented with 1% (*v*/*v*) antibiotic-antimycotic solution (Zell Shield, Minerva Biolabs, Berlin, Germany) and cells were grown at 37 °C in an atmosphere of 5% of CO_2_.

The respiratory syncytial virus (RSV) A2 strain (ATCC^®^ VR-1540), the human rotavirus (HRoV) strain Wa (ATCC^®^ VR-2018), and the herpes simplex virus type 2 (HSV-2) strain MS (ATCC^®^ VR-540) were propagated in Hep2, MA104, and Vero cells, respectively [67,68]. The human influenza virus type A strain A/New Caledonia/20/99 (FluVA-H1N1; Italian National Institute of Health) was propagated in canine kidney epithelial MDCK cells (ATCC^®^ NBL-2) with a culture medium supplemented with 0.5 µg/mL of porcine pancreas trypsin (Sigma-Aldrich; St. Louis, MO, USA) [69]. After propagation, the obtained viral progenies were clarified out of cell debris, aliquoted, and stored at −80 °C prior to virus titration by standard plaque assay or focus reduction assay, dependently on the viral agent considered [67,68].

The antiviral assays were performed with DMEM supplemented with 2% of FBS (for RSV and HSV-2), or without serum (in the case of HRoV and FluVA).

### 3.9. Antibodies and Reagents

The mouse monoclonal antibodies recognizing RSV F protein (ab43812) and FluVA-H1N1 nucleoprotein (ab20343) were purchased from Abcam plc. (Cambridge, UK), while the mouse monoclonal antibody targeting HRoV VP6 protein (mab0036-P) was purchased from Covalab (Bron, France).

Secondary AffiniPure F(ab’)2 Fragment Goat Anti-Mouse IgG (H + L) peroxidase-conjugated antibody was purchased from Jackson ImmunoResearch Europe Ltd. © (115-036-003; St. Thomas’ Place, Ely, UK).

Polymers were dissolved in sterile phosphate-buffered solution (PBS) to a concentration of 10 mg/mL and used for the antiviral assays.

Methylcellulose and crystal violet were purchased from Sigma-Aldrich, while absolute ethanol was from Fisher Chemical (Thermo Fisher Scientific; Waltham, MA, USA).

### 3.10. Virus Inhibition Assays

The antiviral activity of the polymers was determined by focus reduction assay for RSV, HRoV, and FluVA and by plaque reduction assay for HSV-2. Confluent cells in 96-well plates (Hep-2, MA104 and A549) or in 24-well plates (Vero) were pre-treated with serial dilutions of polymers (from 2700 to 0.14 µg/mL unless otherwise stated) for 1 h at 37 °C. Untreated control wells were treated with culture medium supplemented with equal volumes of sterile PBS (from 3% (*v*/*v*) to 0.0014% (*v*/*v*)). Simultaneously, mixtures of serial dilutions of polymers or sterile PBS as control and a fixed viral inoculum (MOIs of 0.01 for RSV, HRoV and H1N1, and MOI of 0.004 for HSV-2) were incubated for 1 h at 37 °C. After a single wash, the mixtures were added to pre-treated cells for 1 h in the case of HRoV and HSV-2, for 2 h in the case of FluVA, and for 3 h in the case of RSV. After two washes, the infected A549 and MA104 cells were incubated with a fresh medium for 16 h or 24 h, respectively, at 37 °C, while the infected Hep-2 and Vero cells were overlaid with a 1.2% methylcellulose medium for 72 h or 24 h, respectively. A549, MA104, and Hep-2 cells were fixed with cold acetone-methanol (50:50) and the FluVA-, HRoV-, and RSV-infected cells were detected by means of indirect immunostaining [67,68]. Immunostained viral foci were microscopically counted and results were reported as percentages of controls. HSV-2-infected cells were fixed and stained with 0.1% crystal violet in 20% ethanol and viral plaques were counted. The mean plaque count for each sample dilution was expressed as a percentage of the mean plaque count of the control.

### 3.11. Viability Assay

Cell viability was assessed using the MTS [3-(4,5-dimethylthiazol-2-yl)-5-(3-carboxymethoxyphenyl)-2-(4-sulfophenyl)-2H-tetrazolium] (Promega; Madison, WI, USA) assay as previously described [70]. Confluent Hep-2, MA104, A549, and Vero cells were treated with serial dilutions of polymers under the same experimental conditions of the virus inhibition assays. The 50% cytotoxic concentration (CC_50_) was determined using the software GraphPad Prism version 8.0 (GraphPad Software, San Diego, CA, USA). Where possible, a selectivity index (SI) was calculated by dividing the CC_50_ by the EC_50_ value.

### 3.12. Binding Assays

Vero and Hep-2 cells were pre-seeded in a 24-well plate. The following day, cells and viruses (HSV-2 or RSV, MOIs of 3) were cooled to 4 °C for 10 min. The viruses were then allowed to attach to the cells in the presence of high concentrations of the selected polymer (≥EC_90_) for 1 h on ice, and cells were then washed twice with a cold medium to remove any unbound virus. Subsequently, cells were collected and subjected to three rounds of freeze–thawing to release any cell-bound virus. The cell lysates were clarified by means of low-speed centrifugation for 10 min, and the cell-bound virus titers were determined by means of plaque assay (HSV-2) or indirect immunostaining (RSV), as outlined above. Titer reduction was assessed by means of a one-way ANOVA statistic test followed by Bonferroni post-test, using the software GraphPad Prism version 8.0 (GraphPad Software, San Diego, CA, USA)

### 3.13. Virus Inactivation Assays

Inactivation assays were performed to evaluate the ability of polymers to directly alter the HSV-2 and RSV particles, thus compromising their penetration inside the host cell. About 5 × 10^5^ PFUs of HSV-2 or RSV were incubated for 2 h at 37 °C together with high concentrations of polymers (≥EC_90_) and then titrated on Vero or Hep-2 cells, respectively, until non-inhibitory dilutions of the compounds by means of tailored standard titration methods, as described above. The residual viral infectivity was assessed by means of one-way ANOVA statistic test followed by Bonferroni post-test, using the software GraphPad Prism version 8.0 (GraphPad Software, San Diego, CA, USA)

### 3.14. Data Analysis

All results are presented as the mean values from three independent experiments performed in duplicate. The EC_50_ values for inhibition curves were calculated by regression analysis using the software GraphPad Prism version 8.0 (GraphPad Software, San Diego, CA, USA) by fitting a variable slope-sigmoidal dose–response curve. Statistical analysis was performed using Student’s test, ANOVA analysis of variance, or the F-test, as reported in the legends of the tables and figures.

## 4. Conclusions

Overall, this study discloses the anti-HSV-2 and anti-RSV activity of a new βCD-based hyper-branched negatively charged polymer, named P_PMDA. By specific assays, we demonstrated that the polymer is endowed with virucidal activity against RSV and with binding inhibitory activity against HSV-2. Our results suggest that P_PMDA can be exploited to produce new antiviral materials to counteract RSV dissemination through air or direct contact or can be a good starting material for the study and development of new therapeutic topical options against genital HSV-2 infection. Further studies are needed to verify the virucidal potential of P_PMDA when employed to produce antiviral materials and to assess its biocompatibility, safety, and antiviral action in in vivo models.

## Figures and Tables

**Figure 1 ijms-23-08701-f001:**
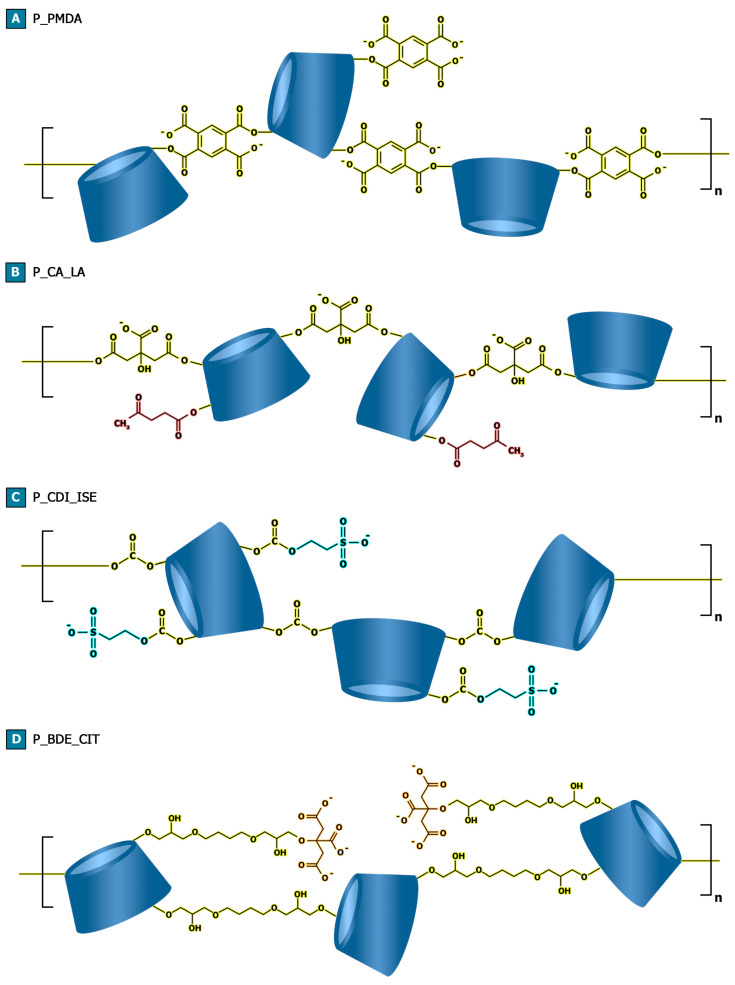
Schematic representation of (**A**) P_PMDA, (**B**) P_CA_LA (**C**) P_CDI_ISE, and (**D**) P_BDE_CIT.

**Figure 2 ijms-23-08701-f002:**
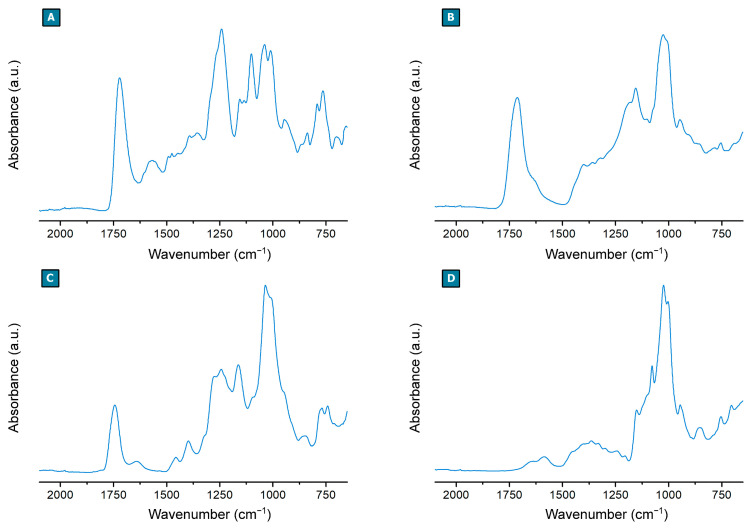
FTIR-ATR spectra of (**A**) P_PMDA, (**B**) P_CA_LA (**C**) P_CDI_ISE, and (**D**) P_BDE_CIT.

**Figure 3 ijms-23-08701-f003:**
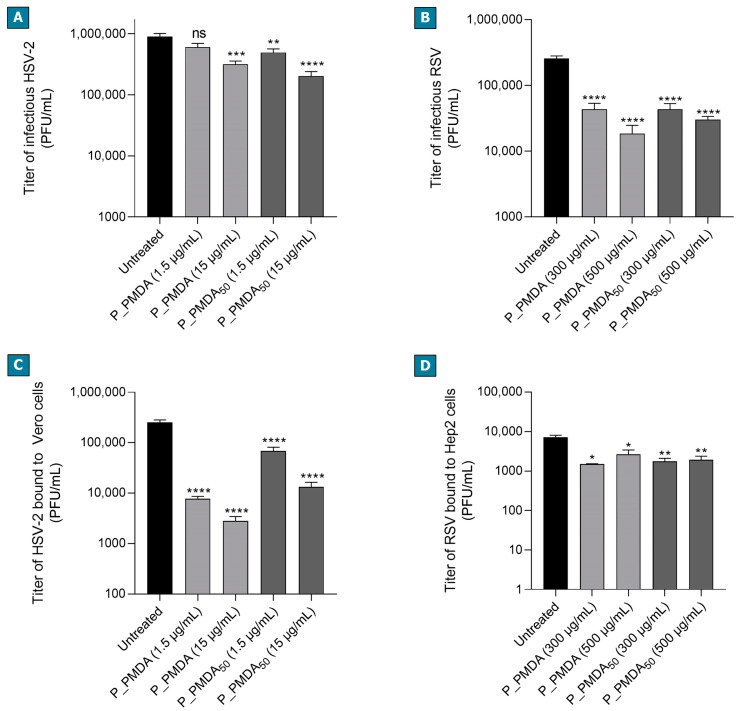
Investigation of the mechanism of action of P_PMDA and its active fraction. HSV-2 (**A**) and RSV (**B**) inactivation assays. HSV-2 or RSV infectious particles were incubated with high concentrations (EC_90_ or higher doses) of P_PMDA or P_PMDA_50_ for 2 h at 37 °C and the residual viral infectivity was then evaluated by titration to the non-inhibitory concentration of the polymers. For the binding assays, HSV-2 (**C**) or RSV (**D**) (MOIs = 3) were allowed to attach to cells in the presence of the same concentrations of polymers reported for the virus inactivation assays. The cell-bound virus titers were determined by means of titration on confluent cells. On the y-axis, the infectious titers are expressed as plaque-forming units per mL (PFU/mL). Error bars represent standard error of the mean (SEM) of three independent experiments (ANOVA and Bonferroni post hoc test; * *p* < 0.05, ** *p* < 0.005, *** *p* < 0.0005, **** *p* < 0.0001, ns: not significant).

**Table 1 ijms-23-08701-t001:** Main characteristics of synthesized polymers.

Polymer	T_onset_ (°C)	ζ-Potential (mV)	Acidity (m_eq_)	Sulfur (wt.%)
P_PMDA	165	−31.4 ± 1.7	872 ± 10	\
P_CA_LA	160	−10.6 ± 1.1	755 ± 9	\
P_CDI_ISE	240	−17.5 ± 0.6	80 ± 9	6.11 ± 0.16
P_BDE_CIT	270	−3.4 ± 1.0	33 ± 7	\

**Table 2 ijms-23-08701-t002:** Antiviral activity of βCD-based polymers against RSV, HSV-2, FluVA, and HRoV.

	Polymer	* EC_50_ (µg/mL) (95% ^§^ CI)	^#^ EC_90_ (µg/mL) (95% CI)	^‡^ CC_50_ (µg/mL) (95% CI)	^ⴕ^ SI
**RSV**	P_PMDA	9.72 (7.38–12.81)	157.6 (82.87–299.6)	>3000	>309
	P_CA_LA	33.29 (19.41–62.87)	1335 (356–9853)	>3000	n.a.
	P_CDI_ISE	326.60 (87.60–6784)	>2700	>3000	n.a.
	P_BDE_CIT	>2700	>2700	n.t.	n.a.
**HSV-2**	P_PMDA	0.18 (0.16–0.2)	0.92 (0.77–1.12)	>3000	>16,667
	P_CA_LA	4.12 (2.18–8.7)	44.06 (5.57–544.4)	>3000	>728
	P_CDI_ISE	>2700	>2700	n.t.	n.a.
	P_BDE_CIT	>2700	>2700	n.t.	n.a.
**FluVA**	P_PMDA	61.40 (41.71–92.84)	545.7 (244.7–1510)	>3000	>49
	P_CA_LA	>2700	>2700	n.t.	n.a.
	P_CDI_ISE	>2700	>2700	n.t.	n.a.
	P_BDE_CIT	>2700	>2700	n.t.	n.a.
**HRoV**	P_PMDA	>2700	>2700	n.t.	n.a.
	P_CA_LA	>2700	>2700	n.t.	n.a.
	P_CDI_ISE	>2700	>2700	n.t.	n.a.
	P_BDE_CIT	>2700	>2700	n.t.	n.a.

* EC_50_: 50% effective concentration; ^§^ CI: confidence interval; ^#^ EC_90_: 90% effective concentration; ^‡^ CC_50_: 50% cytotoxic concentration; ^ⴕ^ SI: selectivity index; n.a. not assessable; n.t. not tested.

**Table 3 ijms-23-08701-t003:** Antiviral activity of P_PMDA and its fractions against RSV and HSV-2.

	Polymer	* EC_50_ (µg/mL) (95% CI ^§^)	^#^ EC_90_ (µg/mL) (95% CI)	^‡^ CC_50_ (µg/mL) (95% CI)	^ⴕ^ SI	** *p*-Value
**RSV**	P_PMDA	9.72 (7.38–12.81)	157.6 (82.87–299.6)	>3000	>309	
	P_PMDA_50_	4.47 (2.70–7.39)	195 (60.57–627.5)	>3000	>671	0.5305
	P_PMDA_30/50_	8.18 (4.69–14.27)	1510 (317–7177)	>3000	>367	0.3638
	P_PMDA_10/30_	52.58 (31.39–88.07)	1031 (255.3–4164)	>3000	>57	<0.0001
**HSV-2**	P_PMDA	0.18 (0.16–0.2)	0.92 (0.77–1.12)	>3000	>16,667	
	P_PMDA_50_	0.09 (0.07–0.13)	0.29 (0.16–0.66)	>3000	>33,333	<0.0001
	P_PMDA_30/50_	0.13 (0.10–0.17)	0.73 (0.47–1.23)	>3000	>23,077	0.0719
	P_PMDA_10/30_	0.33 (0.20–0.51)	7.12 (3.03–21.62)	>3000	>9091	0.0048

* EC_50_: 50% effective concentration; ^§^ CI: confidence interval; ^#^ EC_90_: 90% effective concentration; ^‡^ CC_50_: 50% cytotoxic concentration; ^ⴕ^ SI: selectivity index; ** *p*-value resulting from Fisher’s test comparing EC_50_ value of the single fraction with the one of P_PMDA.

## Data Availability

The data presented in this study are available on request from the corresponding author.

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
