# Peer review of "Identification of a βCD-Based Hyper-Branched Negatively Charged Polymer as HSV-2 and RSV Inhibitor"

_ijms, 2022, doi:10.3390/ijms23158701_

Round 1
Reviewer 1 Report
The manuscript has undergone the suggested changes and it is almost ready for publication. This reviewer would, however, strongly recommend increasing the quality and resolution of Figures 2 and 3.
Please note also that figure 2 has no caption for the axis. This must be corrested. The reviewer is assuming that the xx axis should be wavenumber / cm-1 and that the yy axis should be Absorbance / a.u. (standing for arbitrary units) but this must be clarified by the authors.
Author Response
1) We thank the Reviewer for the suggestion. All the figures are saved and added to the manuscript with 600dpi resolution. However, the .doc and .pdf files provided for the Reviewers have a downgraded image quality. We will kindly ask the editorial office to provide, if possible, a non-downgraded version of the revised manuscript.
2) We agree with the Reviewer. The figure has been modified accordingly.

Reviewer 2 Report
This is an interesting study and the authors have collected a unique dataset using cutting edge methodology. The paper writing is generally improved.
Author Response
We thanks the referee for the positive evaluation of the revised manuscript

This manuscript is a resubmission of an earlier submission. The following is a list of the peer review reports and author responses from that submission.
Round 1
Reviewer 1 Report
The context is need to be sub sectioned. New line of information should start with new paragraph.
Page 2, at the last line mentioned figure S5B, there is no such figure in the manuscript.
Page 3, line 4, line 13, line 17, line 20, line 26 mentioned figures which not exist in manuscript.
Page 4, last line, at least at beginning need to use full name of analatical process that was used.
There is a lot of abbreviation which have not mentioned full name anywhere in the manuscript. (e.g. TGA, DTGA, and FTIR).
Page 7, figures S7A, S8 and tables SI and SII.
Chemical need to be given with full name. e.g. DMSO, TEA, PMDA, AND ……
There is no clear conclusion and what future work need to be done.
Reviewer 2 Report
The article presents very interesting results about a new class of molecules, cyclodextrin polymers. In the present case, these are presented as candidates to antiviral agents and their activity is investigated in vitro. While the work is overall good and presented in an adequate form, some aspects still warrant improvement.
1) The new cyclodextrin molecules/(monomer for each polymer) should be presented in the main text of the article, so that the readers get familiar with their structures. This way, authors should include a figure or scheme with the proposed structures for each of the new four cyclodextrins, and place their acronym under the structure for easy reference.
2) The article has an excessive use of acronyms. Also, the use of two different acronyms for citrate (one for the neutral form and another for the sodium salt) is extremely confusing. Molecules with simple names such as citric acid, sodium citrate and isethionic acid do not need to be turned into acronyms.
3) TGA data does not seem to bring very important information regarding the structure of the new cyclodextrins nor their biological activity, so perhaps it makes more sense to include this information in the supplementary material.
4) Regarding the antiviral activity studies:
4.1) Ideally, concentrations should be expressed in molarity (or subunits) so that the results can be properly compared, because different polymers do not have the same molecular weight. An average molecular weight for each polymer can be used to do this conversion.
4.2) The expression "n.a." (not assessable) in the table II is not informative and it must be removed. If the highest concentration tested was ineffective, authors should present results as ">[highest concentration tested]" (similar to what was done for P-PMDA against FluV A, which shows CC50 > 3000). Concominantely, the range of concentrations tested should be clearly indicated in the experimental section.
5) Regarding safety of the molecules/polymers, the authors claim that the "strong binding inhibitory activity against HSV-2", "water solubility" of P_PMDA and "acyclovir complexation potential of βCD" are good enough characteristics for further development "as a new therapeutic topical option to treat genital HSV-2 infection". This statement is quite exaggerated and it needs revising to include mention to other relevant properties that still need to be studied, such as proof that no unreacted molecules of PMDA are remaining in the polymer (PMDA, like most anhydrides, can conjugate to human proteins to form a chemical-protein complex with immunogenic properties, i.e., an allergen), verification of in vivo activity and overall assessment of biological compatibility and toxicological safety.
6) Finally, a minor correction:
Section 2.2., first line: please change "with the purpose to synthesize" to "with the purpose of synthesizing"